# Structural insight into TPX2-stimulated microtubule assembly

**Rui Zhang[1†‡], Johanna Roostalu[2†], Thomas Surrey[2]\*, Eva Nogales[1,3,4]\***

[1]Molecular Biophysics and Integrative Bioimaging Division, Lawrence Berkeley National Laboratory, Berkeley, United States; [2]The Francis Crick Institute, London, United Kingdom; [3]Department of Molecular and Cell Biology, University of California, Berkeley, Berkeley, United States; [4]Howard Hughes Medical Institute, University of California, Berkeley, Berkeley, United States

**\*For correspondence:**
Thomas.Surrey@crick.ac.uk (TS);
enogales@lbl.gov (EN)

[†]These authors contributed equally to this work

**Present address:** [‡]Department of Biochemistry and Molecular Biophysics, Washington University School of Medicine, St. Louis, United States

**Competing interests:** The authors declare that no competing interests exist.

**Abstract** During mitosis and meiosis, microtubule (MT) assembly is locally upregulated by the chromatin-dependent Ran-GTP pathway. One of its key targets is the MT-associated spindle assembly factor TPX2. The molecular mechanism of how TPX2 stimulates MT assembly remains unknown because structural information about the interaction of TPX2 with MTs is lacking. Here, we determine the cryo-electron microscopy structure of a central region of TPX2 bound to the MT surface. TPX2 uses two flexibly linked elements ('ridge' and 'wedge') in a novel interaction mode to simultaneously bind across longitudinal and lateral tubulin interfaces. These MT-interacting elements overlap with the binding site of importins on TPX2. Fluorescence microscopy-based in vitro reconstitution assays reveal that this interaction mode is critical for MT binding and facilitates MT nucleation. Together, our results suggest a molecular mechanism of how the Ran-GTP gradient can regulate TPX2-dependent MT formation.
DOI: https://doi.org/10.7554/eLife.30959.001

## Introduction

The microtubule (MT) cytoskeleton is essential for correct intracellular organization, cell division and differentiation. MT function depends on a variety of MT-associated proteins (MAPs) that control MT nucleation, dynamics and interactions with other cellular structures. Among them is TPX2 (*Wittmann et al., 1998*), a MAP from multicellular eukaryotes that is nuclear during interphase (*Neumayer et al., 2012*) and associates with spindle MTs after nuclear breakdown during mitosis and meiosis (*Garrett et al., 2002*; *Gruss et al., 2002*; *Heidebrecht et al., 1997*; *Neumayer et al., 2014*). TPX2 is a multifunctional protein with several mitotic/meiotic activities (*Neumayer et al., 2014*). Both over- and under-expression of TPX2 perturb MT organization, leading to genomic instability, and mutations in TPX2 are correlated with high metastasis frequency in cancer patients (*Aguirre-Portolés et al., 2012*; *Carter et al., 2006*; *Gruss et al., 2002*; *Pérez de Castro and Malumbres, 2012*). Consequently, TPX2 is a marker for the diagnosis and prognosis of malignancies (*Gruss et al., 2002*; *Heidebrecht et al., 1997*; *Neumayer et al., 2014*).

TPX2 is a critical component of the so-called Ran-pathway (*Cavazza and Vernos, 2015*). Local production of Ran-GTP around mitotic and meiotic chromosomes liberates proteins that contain a nuclear localization signal (NLS), including a set of spindle assembly factors, from the inhibitory action of importins (nuclear transport receptors). TPX2 is one such spindle assembly factor, having prominent roles in local MT formation around chromatin (*Gruss et al., 2001*; *Gruss et al., 2002*; *Petry et al., 2013*) and targeting of other spindle components to the spindle (*Cavazza and Vernos, 2015*). The molecular mechanism and the control of TPX2-dependent MT nucleation is still poorly understood.

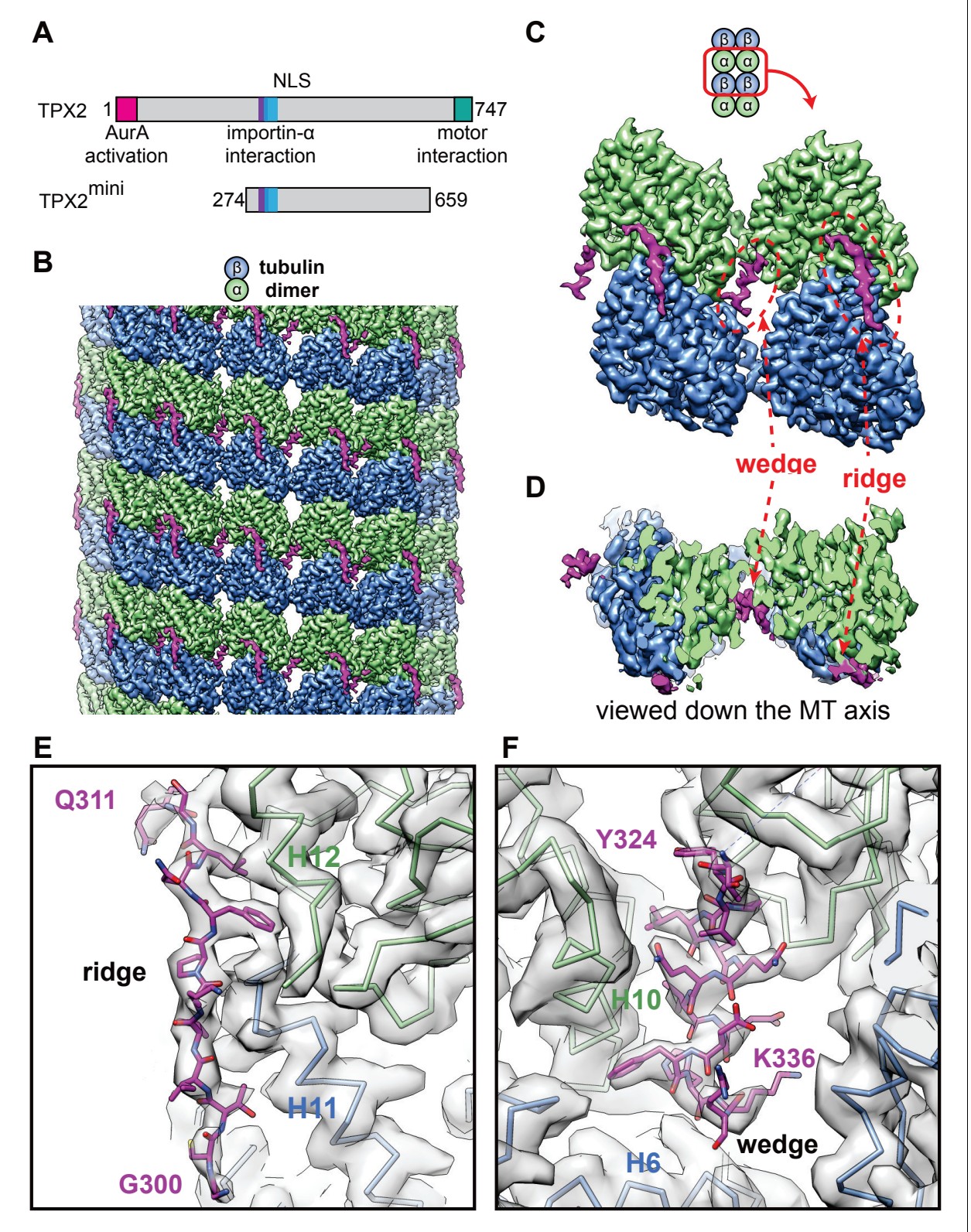

**Figure 1.** High-resolution cryo-EM structure of TPX2 bound to GMPCPP-MTs. (A) Schematic of domain structure for full-length TPX2 and TPX2$^{mini}$. (B) Cryo-EM reconstruction of mGFP-TPX2$^{mini}$ decorated GMPCPP-MT, with pseudo-helical symmetry applied. α-tubulin, β-tubulin and TPX2 are colored in green, blue and magenta, respectively. The same color scheme is used throughout. (C) Zoom-in view of two TPX2 molecules interacting with four

*Figure 1 continued on next page*

*Figure 1 continued*

neighboring tubulin monomers. (**D**) End-on view of the cryo-EM density, related to (c) by a 90° rotation, looking toward the MT minus end.( **E–F**) Zoom-in view of the cryo-EM density and atomic model of TPX2 ridge (**E**) and wedge (**F**) respectively.

DOI: https://doi.org/10.7554/eLife.30959.002

The following figure supplements are available for figure 1:

**Figure supplement 1.** The sequence of human TPX2.

DOI: https://doi.org/10.7554/eLife.30959.003

**Figure supplement 2.** Coomassie Blue stained SDS page gels of purified TPX2 constructs used in this study.

DOI: https://doi.org/10.7554/eLife.30959.004

**Figure supplement 3.** Cryo-EM images of GMPCPP-MTs decorated with TPX2^mini or TPX2^micro constructs.

DOI: https://doi.org/10.7554/eLife.30959.005

**Figure supplement 4.** Resolution estimation of the cryo-EM structures of GMPCPP-MTs decorated with TPX2.

DOI: https://doi.org/10.7554/eLife.30959.006

**Figure supplement 5.** Cryo-EM reconstructions of GMPCPP-MTs decorated with TPX2 molecules.

DOI: https://doi.org/10.7554/eLife.30959.007

TPX2 is an elongated monomeric protein composed of several functionally distinct parts (*Figure 1A*, *Figure 1—figure supplement 1*). In addition to several MT-binding regions (*Alfaro-Aco et al., 2017*; *Brunet et al., 2004*; *Roostalu et al., 2015*; *Trieselmann et al., 2003*), TPX2 interacts with numerous binding partners. Its N-terminus interacts directly with Aurora A kinase, thereby activating and targeting Aurora A to the spindle (*Bayliss et al., 2003*; *Eyers et al., 2003*; *Giubettini et al., 2011*; *Kufer et al., 2002*; *Tsai et al., 2003*). TPX2-stimulated Aurora A activity is important for proper spindle assembly, centrosome function and γ-tubulin ring complex (γ-TuRC) activation (*Pinyol et al., 2013*; *Scrofani et al., 2015*; *Tsai and Zheng, 2005*).

The C-terminal half of TPX2 contains α-helical repeats that were shown to be important for stimulating augmin-mediated branching MT nucleation in *Xenopus laevis* egg extract (*Alfaro-Aco et al., 2017*; *Sanchez-Pulido et al., 2016*). The very C-terminus of TPX2 interacts with the two mitotic kinesins Kif11/Eg5 and Kif15/Xklp2 (originally having given the protein its name: TPX2 is short for 'targeting protein for Xklp2') (*Eckerdt et al., 2008*; *Ma et al., 2010*; *Tanenbaum et al., 2009*; *Wittmann et al., 2000*), mediating proper spindle localization of these motors (*Helmke and Heald, 2014*; *Ma et al., 2010*; *Wittmann et al., 1998*; *Wittmann et al., 2000*). TPX2 has also been observed in complex with other spindle-associated MAPs (*Koffa et al., 2006*), the functional significance of which is not well understood.

In vitro experiments with purified proteins have demonstrated that TPX2 directly promotes MT stability by reducing the frequency of catastrophes (transition from MT growth to depolymerization) and by slowing down depolymerization (*Reid et al., 2016*; *Roostalu et al., 2015*; *Wieczorek et al., 2015*). Furthermore, TPX2 can directly stabilize MT nucleation intermediates (*Roostalu et al., 2015*) thereby efficiently stimulating MT nucleation in pure tubulin solutions (*Roostalu et al., 2015*; *Wieczorek et al., 2015*; *Woodruff et al., 2017*). The central portion of TPX2 also contains the NLS whose interaction with importin-α is well characterized, both biochemically and at a structural level (*Giesecke and Stewart, 2010*; *Schatz et al., 2003*). Importins suppress TPX2 binding to MTs and MT nucleation in vitro (*Roostalu et al., 2015*; *Schatz et al., 2003*). The structural basis of the effects of TPX2 on MT stabilization and nucleation as well as their regulation by importins is not yet understood.

In in vitro experiments with purified proteins, TPX2 bound with higher affinity to growing MT ends than to the rest of the MT, a preference likely resulting from sensitivity to the characteristic nucleotide state and/or the curvature of the MT surface at MT ends (*Roostalu et al., 2015*). Accordingly, TPX2 binds also with increased affinity to MTs grown in the presence of the non-hydrolyzable GTP analog GMPCPP (*Roostalu et al., 2015*), a nucleotide which is well known to stabilize MTs and to efficiently promote MT nucleation (*Hyman et al., 1992*). Cryo-electron microscopy (cryo-EM) studies revealed that GMPCPP-MTs have a more extended lattice structure with a slightly different lattice twist compared to GDP-MTs, a conformational difference thought to reflect the more stable GTP state of the MT (*Alushin et al., 2014*; *Hyman et al., 1992*; *Zhang et al., 2015*). The central TPX2 fragment (residues 274–659, *Figure 1A*) was shown to be sufficient for this nucleotide and curvature-sensitive MT binding, albeit with reduced affinity, and for stimulating MT nucleation, even if

to a lesser extent than the full-length protein (*Roostalu et al., 2015*). This raises the possibility that the nucleotide sensitivity of TPX2 and its direct effects on MT dynamics and nucleation might be linked and that they are encoded in the central part of the molecule.

To better understand the molecular mechanism of the effects of TPX2 on MT nucleation and dynamics, we used cryo-EM to determine the atomic structure of TPX2 bound to GMPCPP-MTs. We observed a novel MT-binding mode with two flexibly linked elements of TPX2 binding the outer MT surface, across both longitudinal and lateral tubulin dimer interfaces. The MT-binding region of TPX2 directly overlaps with the NLS and the importin-α interaction motif. The structural results were further validated by mutational analysis and in vitro total internal reflection fluorescence microscopy (TIRFM) assays. The novel MT-binding mode provides a structural explanation for how TPX2 suppresses MT dynamics and stimulates MT nucleation, and how the Ran-GTP gradient can regulate TPX2-MT interaction through importins.

## Results

### TPX2 has a unique MT-binding mode across several interfaces

We used high-resolution cryo-EM to visualize the interaction of the central part of TPX2 (residues 274–659, called TPX2$^{mini}$) (*Figure 1A*, *Figure 1—figure supplements 1* and *2B*) (*Roostalu et al., 2015*) with GMPCPP-MTs. This construct maintains the binding specificity of full-length TPX2, despite its reduced affinity and is amenable to structural studies, because it does not induce MT bundling at the high protein concentration (μM) typically required for cryo-EM studies of MAPs, in contrast to full-length TPX2 (*Brunet et al., 2004*; *Schatz et al., 2003*). Using a MT seam search protocol that allows the structural study of MAPs with relatively small footprints on the MT lattice (*Zhang and Nogales, 2015*), we obtained a 3.3 Å resolution reconstruction of mGFP-TPX2$^{mini}$ decorated MTs (*Figure 1B*, *Figure 1—figure supplements 3A* and *4*).

The structure shows a repeating unit of two small and discontinuous densities on the MT surface corresponding to the TPX2$^{mini}$ molecule (*Figure 1B*), which is predicted to be largely intrinsically disordered (*Figure 1—figure supplement 1*). We refer to these two densities as the 'ridge' and the' wedge' (*Figure 1C*). The ridge binds on the crest of the protofilament (PF) with an extended conformation, while the wedge corresponds to a short α-helix that binds within the crevasse between two adjacent PFs and appears to be 'wedging' between neighboring tubulin subunits (*Figure 1D*). The absence of a connection between these two regions of density indicates that the linker between them is flexible and not in a fixed position with respect to the MT. The ridge is oriented roughly along the MT axis and interacts with both α- and β-tubulin across a longitudinal inter-dimer interface. The half-buried wedge between adjacent PFs interacts also with α- and β-tubulin in one PF, and with another α-tubulin in the neighboring PF. Therefore, both regions of TPX2 bind over tubulin polymerization interfaces, suggesting a potential explanation for how TPX2 can stabilize MTs and stimulate MT nucleation. Thus, TPX2 uses a novel mode of MAP-MT interaction that involves two structural elements, connected by a flexible linker, to simultaneously interact across both longitudinal and lateral tubulin dimer interfaces in the lattice.

The modular MT engagement by TPX2 is different from previously characterized MAP-MT interactions, which typically involve a globular domain (*Nogales and Zhang, 2016*). Using two separate elements connected by a linker that allows for some flexibility, likely also explains why TPX2 binding is not sensitive to the exact PF number of MTs. We observed similar binding to GMPCPP-MTs with either 13 or 14 PFs, which are the most typical PF numbers for these MTs polymerized in vitro (*Figure 1—figure supplement 5A,B*). This is in contrast to end binding proteins of the EB1 family (EBs) and to the MT-stabilizing protein doublecortin (DCX), both of which use compact globular non-flexible domains to 'staple' across two adjacent PFs, resulting in a marked preference for 13-PF MTs (*Fourniol et al., 2010*; *Maurer et al., 2012*; *Zhang et al., 2015*), the typical PF number found in most cells.

### Atomic model of the TPX2-MT interaction

At 3.3 Å resolution, we were able to do de novo modeling of the two MT-binding elements within TPX2, aided by both the position of large side chain densities (*Figure 1E,F*) and secondary structure prediction (*Figure 1—figure supplement 1*). We concluded that the ridge corresponds to residues

300–311, while the wedge corresponds to residues 323–341. These regions overlap extensively with the importin-α-binding site and the NLS of TPX2 (*Giesecke and Stewart, 2010*; *Schatz et al., 2003*) (*Figure 1—figure supplement 1*), directly providing a structural explanation of the inhibitory effects of importins on TPX2 (Discussion).

To test our atomic model of the MT-binding elements within TPX2, we generated a much shorter construct, which we refer to as mGFP-TPX2$^{micro}$. It comprises residues 274 to 370, which more closely encompass the ridge and the wedge elements (*Figure 2A*, *Figure 1—figure supplements 1* and *2A*). TIRFM-based in vitro experiments with purified mGFP-TPX2$^{micro}$ and surface attached GMPCPP-MT 'seeds' from which MTs elongated in the presence of GTP, demonstrated that the shorter TPX2$^{micro}$ construct retains a strong binding preference for GMPCPP-MT 'seeds' (*Figure 2B, C*), like the longer TPX2$^{mini}$, although the overall binding affinity was reduced. Cryo-EM analysis of GMPCPP-MT decorated with the mGFP-TPX2$^{micro}$ construct (*Figure 1—figure supplement 3B*) resulted in a reconstruction with practically identical features to the structure obtained with the longer mGFP-TPX2$^{mini}$ construct (*Figure 2D*), in support of our atomic model for the ridge and the wedge elements.

Based on our atomic model, the ridge and the wedge are connected across adjacent PFs by a short flexible linker of 12 residues. The distance that this linker could stretch roughly matches the distance between the cryo-EM densities of the ridge and the wedge, but is incompatible with simultaneous binding of the two elements across the MT seam, the lattice discontinuity where α-tubulins laterally contact β-tubulins, in contrast to the rest of the lattice. Indeed, asymmetric (C1) reconstruction of the MT confirmed the absence of either the wedge element (for TPX2$^{mini}$) or both elements (for TPX2$^{micro}$) at the MT seam (*Figure 1—figure supplement 5C,D*), in further agreement with our atomic model.

## Confirmation of the TPX2 MT-binding mode using single point mutations

In our atomic model, the ridge of TPX2 binds the MT at the junction between the α:H12 and β:H11 helices in two adjacent dimers (*Figure 3A*). The ridge inserts a phenylalanine F307 into a hydrophobic pocket in tubulin defined by residues α:V435, α:Y262, α:W346 and β:R401 (*Figure 3A*). The wedge of TPX2 interacts simultaneously with the α:H9, α:H10 and β:H6 helices in one PF, and α:H3 and α:H4 helices in the other PF (*Figure 3A*). At the interaction site with the first PF, near the inter-dimer interface, two residues within the wedge, F334 and H335, appear to be critical for TPX2-MT interaction. Interestingly, these identified TPX2 residues are conserved in organisms where TPX2 has been demonstrated to be involved in chromatin-mediated, that is Ran-GTP-dependent, MT nucleation (human, *X. laevis*, *A. thaliana* [*Gruss et al., 2001*; *Gruss et al., 2002*; *Petrovská et al., 2013*; *Vos et al., 2008*]), whereas they are not conserved in organisms where TPX2 appears to play no or only a minor role in this nucleation pathway, despite other involvement in correct spindle assembly (*D. melanogaster*, *C. elegans* [*Hayward et al., 2014*; *Karsenti, 2005*; *Ozlü et al., 2005*]) (*Figure 3—figure supplement 1*).

To further test our atomic model and to determine the importance of the ridge and wedge for overall TPX2 binding, we mutated the three identified residues of mGFP-TPX2$^{micro}$ that are located right at the interfaces between tubulin dimers, and assessed their binding to MTs using a TIRFM-based in vitro assay. We produced two single-residue mutants within the ridge, F307A and F307E, and two double mutants within the wedge, F334A H335A and F334E H335E, as well as the combined triple mutants, F307A F334A H335A and F307E F334E H335E (*Figure 3B*, *Figure 1—figure supplement 2A*). F to A mutations were intended to replace a large charged residue by a small hydrophobic residue, while F to E mutations had the goal to introduce a more drastic change to a negatively charged residue. A similar reasoning applied to the histidine mutations.

We observed that compared to wild-type mGFP-TPX2$^{micro}$ (*Figure 3C*), alanine replacements in either the ridge or the wedge region dramatically reduced the binding strength to GMPCPP-MT seeds (*Figure 3D*, top and middle, respectively, 3F). Simultaneous alanine replacements in both binding regions further reduced the MT interaction of the mutated TPX2$^{micro}$ (*Figure 3D*, bottom, 3F). The corresponding glutamate replacements showed an even more pronounced reduction in the binding affinity (*Figure 3E,F*). Disrupting both the ridge and the wedge by glutamate replacements (F307E F334E H335E mutant) completely abolished all detectable MT binding (*Figure 3E*, bottom,

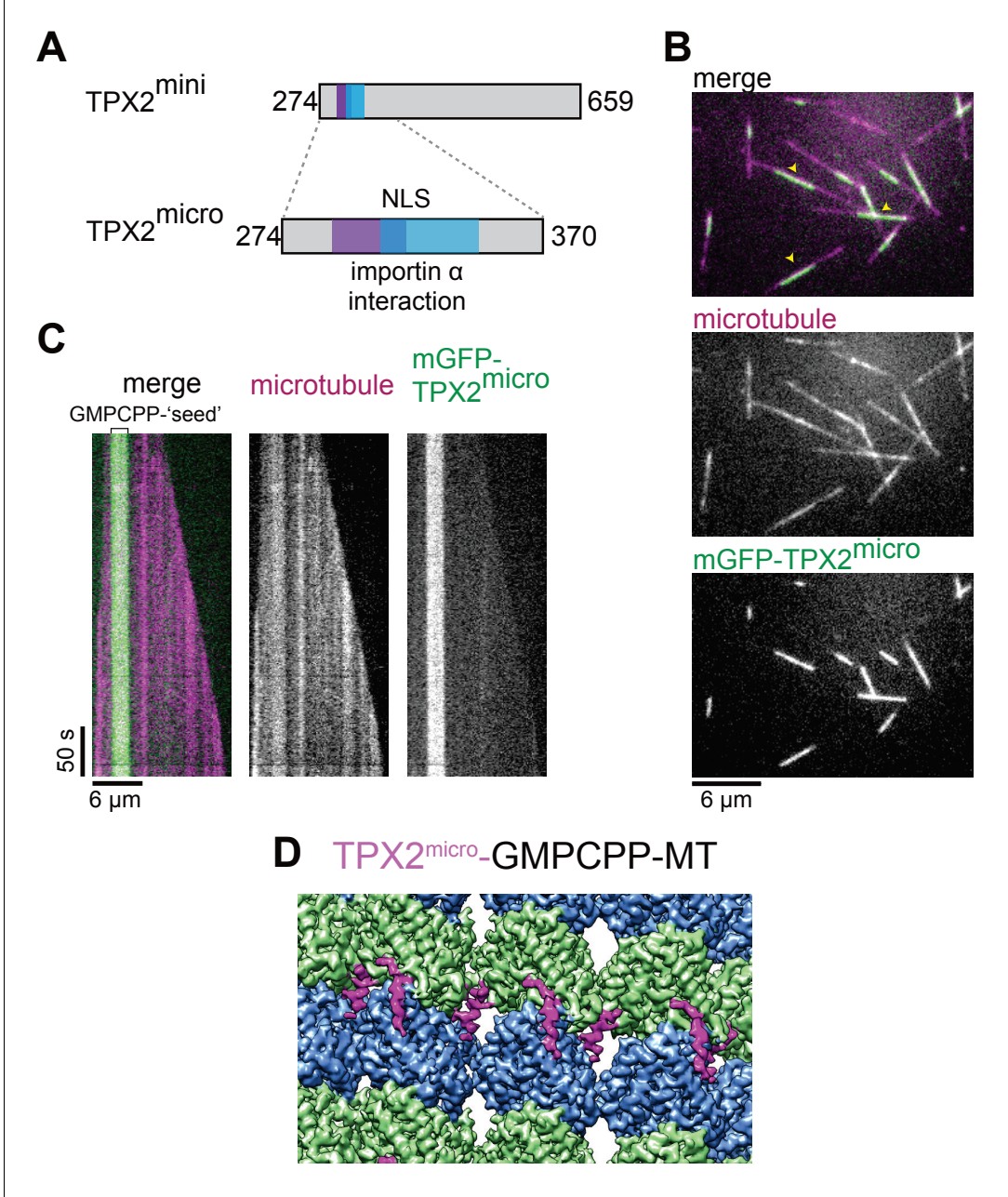

**Figure 2.** TPX2^micro retains the MT lattice specificity for GMPCPP-MTs. (A) Schematic of the TPX2^mini and TPX2^micro constructs. (B–C) TIRF microscopy images (B) and representative kymographs (C) showing mGFP-TPX2^micro (green in merge) binding preferentially to the GMPCPP segment of MTs (magenta in merge) growing dynamically in the presence of GTP. Yellow arrowheads indicate GMPCPP-'seed' region. Tubulin and mGFP-TPX2^micro concentrations were 15 μM and 500 nM, respectively. (D) Reconstruction of GMPCPP-MTs decorated with mGFP-TPX2^micro.

DOI: https://doi.org/10.7554/eLife.30959.008

3F). These data suggest that both the ridge and the wedge are equally important for MT binding of TPX2 and strongly support the validity of the atomic model of the TPX2 MT-binding site.

TPX2^micro in contrast to TPX2^mini did not detectably bind to growing MT ends, probably due to its weaker overall affinity, raising the question of whether the ridge and wedge are critical for growing MT end binding in the context of a longer TPX2 construct. To answer this question, we generated a mGFP-TPX2^mini triple mutant (F307E F334E H335E) with the aim to disrupt ridge and wedge binding also in this longer construct (*Figure 4A* top, *Figure 1—figure supplement 2B*). TIRFM

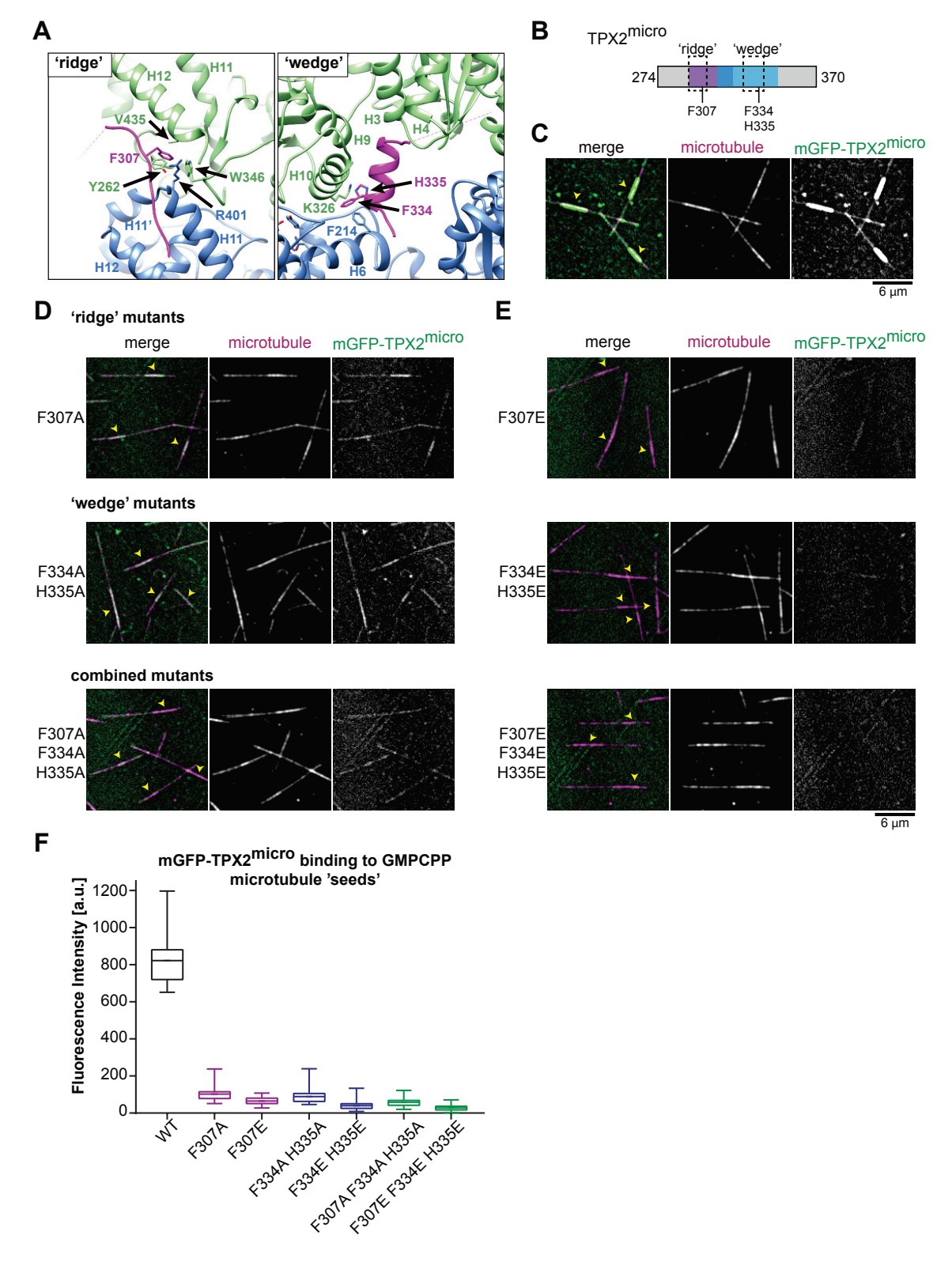

**Figure 3.** Residues important for TPX2 interaction with MTs. (**A**) Zoom-in view of the atomic model of the TPX2 ridge (left) and wedge (right). (**B**) Schematic of TPX2^micro indicating the 'ridge' and the 'wedge' regions as well as the residues that were mutated to test for MT interaction. (**C**) TIRFM images depicting mGFP-TPX2^micro (green) binding to growing Atto565-labeled MTs (magenta). Yellow arrowheads indicate the GMPCPP-'seed' region. (**D–E**) TIRFM images depicting mutant mGFP-TPX2^micro (green) binding to MTs (magenta). Tubulin and mGFP-TPX2^micro concentrations were 15 µM and

*Figure 3 continued on next page*

*Figure 3 continued*

1 µM, respectively. Note that in all cases background subtracted 25-frame averages are shown to allow visualization of the differences between the faint signals of the mutants on the MT lattice. (F) Box-and-whiskers plot depicting average fluorescence intensity measurements for mGFP-TPX2micro GMPCPP MT 'seed' binding comparing wild-type and mutant proteins. The boxes extend from 25th to 75th percentiles, the whiskers extend from minimum to maximum values, and the mean value is plotted as a line in the middle of the box. 500 timeframes were averaged for each MT 'seed'. Number of 'seeds' analyzed: WT – 18, F307A – 30, F307E – 25, F334A H335A – 25, F334E H334 – 29, F307A F334A H335E – 29, F307E F334E H335E – 24.

DOI: https://doi.org/10.7554/eLife.30959.009

The following figure supplement is available for figure 3:

**Figure supplement 1.** Alignment of TPX2 amino acid sequences from different species.

DOI: https://doi.org/10.7554/eLife.30959.010

assays with purified wild-type and mutant TPX2mini (*Figure 4A,B*) indeed demonstrated that binding was abrogated both to GMPCPP-MT seeds (*Figure 4C*) and also to growing MT ends (*Figure 4D*), further confirming that the novel binding module of TPX2 identified here is indeed critical for TPX2 binding to MT ends (see Discussion).

To test the functional significance of the identified binding module, we performed TIRFM-based MT nucleation experiments as described previously (*Roostalu et al., 2015*). In these experiments, MTs nucleate in solution in the presence of biotinylated TPX2 and then bind to a neutravidin-functionalized glass surface (*Figure 4E*). We produced biotinylated constructs of wild-type TPX2mini and the respective triple mutant (*Figure 1—figure supplement 2C*) and compared their efficiency in promoting MT nucleation. Time lapse imaging revealed that the wild-type protein strongly promoted MT nucleation, as expected (*Roostalu et al., 2015*), whereas the MT nucleation ability of the triple mutant was severely compromised (*Figure 4F*). This demonstrates that the identified interaction module is an important facilitator of TPX2-dependent MT formation.

## MT stabilization by TPX2

TPX2 binds next to structural elements within tubulin that show significant local rearrangement during the lattice compaction that accompanies GTP hydrolysis (*Figure 5*), such as the T5 loop in β-tubulin and the H5 helix in α-tubulin (black dashed circles in *Figure 5B,C*) (*Zhang et al., 2015*). Furthermore, comparison of the cryo-EM reconstructions of GMPCPP-MTs in the absence and presence of TPX2mini shows that binding of TPX2mini has a direct effect on the MT lattice structure. TPX2 binding increases slightly the right-handed twist in the lattice (measured as a 'dimer twist'), as well as the inter-dimer distance (measured as 'dimer rise') (*Figure 5—figure supplement 1*, *Figure 5—figure supplement 1—source data 1*). This means that TPX2 appears to oppose the compaction of the MT lattice thought to occur upon GTP hydrolysis (*Alushin et al., 2014*; *Hyman et al., 1992*). The helical character of the wedge of TPX2 may provide sufficient mechanical strength to counteract the decrease in distance between the two wedge-binding sites (red dashed circles in *Figure 5B*) associated with GTP hydrolysis and lattice compaction. Moreover, the binding of TPX2 may lock rotamer conformations of key tubulin residues that contribute to the allosteric response to GTP hydrolysis within the tightly packed space of the MT lattice (*Figure 5—figure supplement 2*). These observations strongly suggest that TPX2 does not only sense, but it also influences the conformational state of the MT, suggesting a structural explanation for the MT stabilizing effect of TPX2.

## Discussion

### A new MT-binding mode

Our high-resolution cryo-EM structure of TPX2 bound to GMPCPP-MTs reveals a new mode of MAP-MT interaction that uses two small structural elements, the wedge and the ridge, connected by a flexible linker, to interact across longitudinal and lateral tubulin dimer interfaces (*Figure 6*). Although the TPX2 wedge binds to a similar location on the MT surface as EB1 and doublecortin (DCX) (between tubulin dimers and between PFs), they interact with different sets of tubulin residues (*Figure 6—figure supplement 1*). The modular MT engagement by TPX2 via extended regions of the protein is in stark contrast to the interaction of globular protein domains with a MT so far

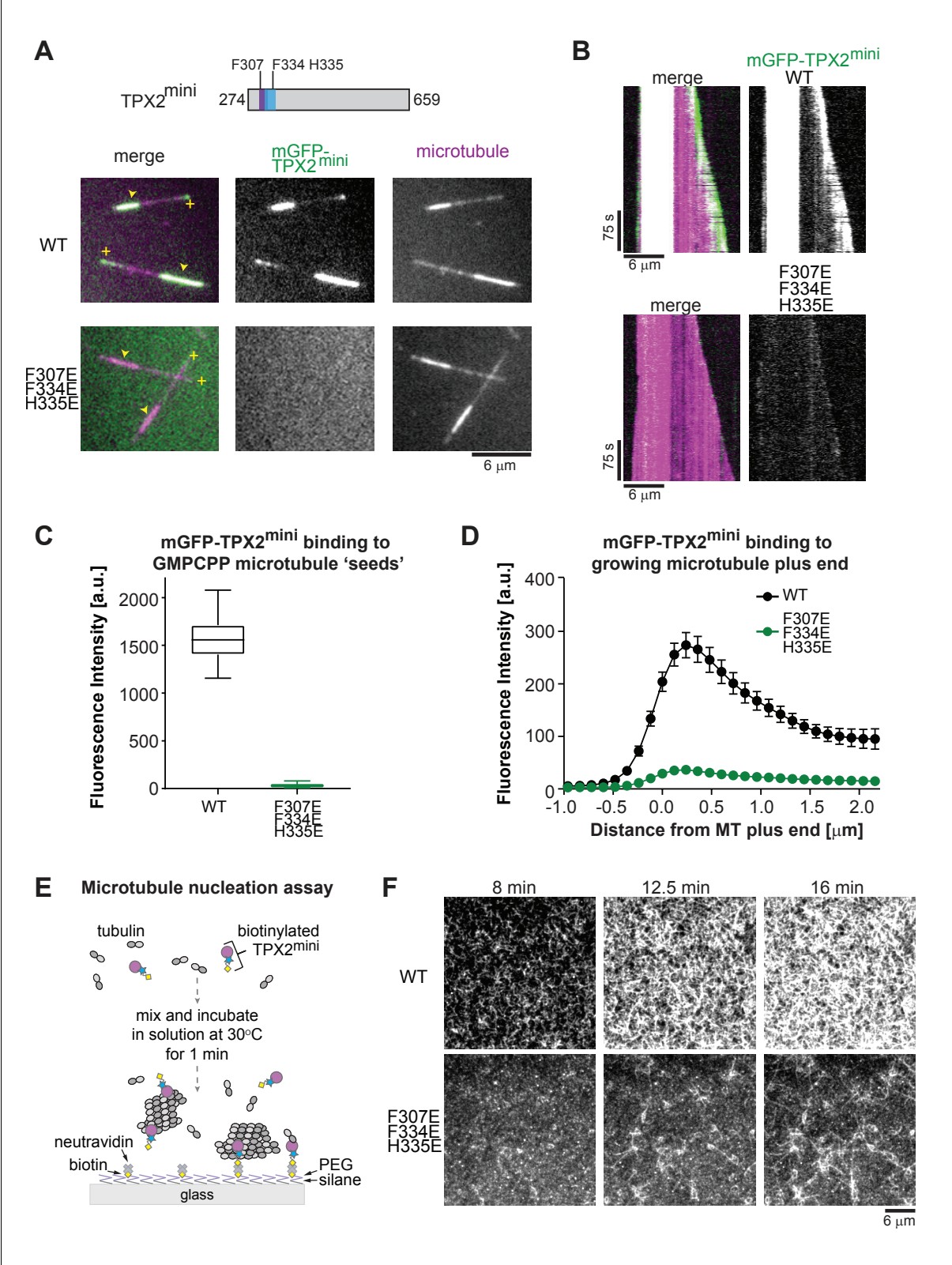

**Figure 4.** Perturbing critical residues for MT interaction disrupts GMPCPP 'seed' and growing MT end binding of TPX2^mini. (**A**) Schematic of the TPX2^mini indicating the three mutated residues (top), and representative TIRF microscopy images (bottom) comparing wild-type mGFP-TPX2^mini and the F307E F334E H335E triple mutant of mGFP-TPX2^mini (green in merge) binding to dynamic Alexa647-labeled MTs (magenta in merge). MT plus ends are indicated by yellow (+) signs, and GMPCPP 'seeds' by yellow arrowheads. (**B**) Representative kymographs of the same experiment. Tubulin and mGFP-

*Figure 4 continued on next page*

*Figure 4 continued*

TPX2^mini concentrations were 12.5 μM and 125 nM, respectively. (**C**) Box-and-whiskers plot depicting average fluorescence intensity measurements for mGFP-TPX2^mini GMPCPP MT 'seed' binding comparing wild-type and mutant protein. The boxes extend from 25^th to 75^th percentiles, the whiskers extend from minimum to maximum values, and the mean value is plotted as a line in the middle of the box. 250 time frames were averaged for each MT 'seed'. Number of 'seeds' analyzed: WT – 57, F307E F334E H335E – 67. (**D**) Averaged fluorescence intensity profiles of wild-type and mutant mGFP-TPX2^mini at growing MT ends. 180 timeframes were averaged for each growing plus end. Number of plus ends analyzed: WT – 42, F307E F335E H335E – 43. Error bars are s.e.m. (**E**) Schematic of the TIRF microscopy-based MT nucleation assay. (**F**) Representative TIRFM images of CF640R-labeled MT nucleation time course comparing the nucleation promoting ability of wild-type biotinylated TPX2^mini and F307E F334E H335E triple mutant biotinylated TPX2^mini. Fluorescently labeled tubulin concentration was 15 μM, biotinylated TPX2^mini concentrations were 90 nM.

DOI: https://doi.org/10.7554/eLife.30959.011

structurally characterized for other MAPs, for example the calponin homology domains of Ndc80 or EB3/Mal3 (*Alushin et al., 2010*; *Maurer et al., 2012*; *Zhang et al., 2015*), the doublecortin domain (*Moores et al., 2004*), kinesin-motor domains (*Goulet et al., 2014*; *Shang et al., 2014*; *Sosa and Milligan, 1996*) or the spectrin domain of PRC1 (*Kellogg et al., 2016*).

This novel TPX2 interaction mode permits bridging of adjacent PFs without the need for a rigidly fixed curvature between them, so that tubulin assembly can be promoted even before MT tube closure occurs at the growing MT end. At the same time, the new binding mode is sensitive to the compaction state of the MT lattice and may even influence the GTP hydrolysis process and/or the allosteric response to GTP hydrolysis.

## MT stabilization by TPX2

Recent in vitro studies have shown that TPX2 suppresses MT dynamics by reducing the frequency of catastrophes and the rate of MT depolymerization (*Reid et al., 2016*; *Roostalu et al., 2015*; *Wieczorek et al., 2015*). Furthermore, it was observed that TPX2 promotes the elongation of the region it binds to at growing MT ends (*Roostalu et al., 2015*), indicating that it affects conformational changes of the MT lattice, potentially suggesting that it slows down GTP hydrolysis. This agrees with our model that TPX2 antagonizes MT lattice compaction (*Figure 6*). Remarkably, end binding proteins (EBs) have opposite effects on the MT lattice parameters (in terms of both lattice spacing and lattice twist) (*Figure 5—figure supplement 1*) (*Zhang et al., 2015*), which is paralleled by opposite effects on the kinetics of GTPase reactions (*Aguirre-Portolés et al., 2012*; *Maurer et al., 2011*; *Maurer et al., 2014*) and on the catastrophe frequency and nucleation efficiency of MTs as compared to TPX2 (*Bieling et al., 2007*; *Komarova et al., 2009*; *Maurer et al., 2014*; *Roostalu et al., 2015*; *Vitre et al., 2008*; *Wieczorek et al., 2015*). Future studies with other MAPs will further test if this applies as a general rule for the relationship between MT lattice parameters and MT stability.

In addition to the stably bound wedge and ridge elements that we see in our cryo-EM structures, TPX2 contains other MT-binding regions that can increase its MT-binding affinity and may additionally contribute to its ability to stabilize MTs (*Alfaro-Aco et al., 2017*; *Brunet et al., 2004*; *Roostalu et al., 2015*; *Trieselmann et al., 2003*). Such additional contacts may shield the electrostatic repulsion between tubulin subunits due to their negatively charged C-terminal tails, and thus also add to the MT stabilization effect of TPX2.

## Stimulation of MT nucleation by TPX2

It has been shown that TPX2 can directly promote MT nucleation in vitro, by stabilizing early nucleation intermediates (*Roostalu et al., 2015*). While the structural nature of these intermediates is poorly understood, we hypothesize that they may share some common features with MT ends, possibly resembling a curved, GTP-rich, sheet-like structure (*Chrétien et al., 1995*; *Guesdon et al., 2016*; *Voter and Erickson, 1984*; *Wang et al., 2005*) (*Figure 6*). By slowing down GTP hydrolysis and therefore the premature transition to an unstable GDP state that is prone to depolymerization, TPX2 would allow the early nucleation intermediates to have more time to grow and transform into elongating MTs (*Roostalu et al., 2015*). In addition, given the flexible character of the linker between its two MT-binding elements, TPX2 may promote the formation of nucleation intermediates by bridging two short PFs and facilitating their lateral association, even before the final lateral curvature between PFs is established within the closed, cylindrical MT lattice (*Figure 6*). Furthermore, given that the

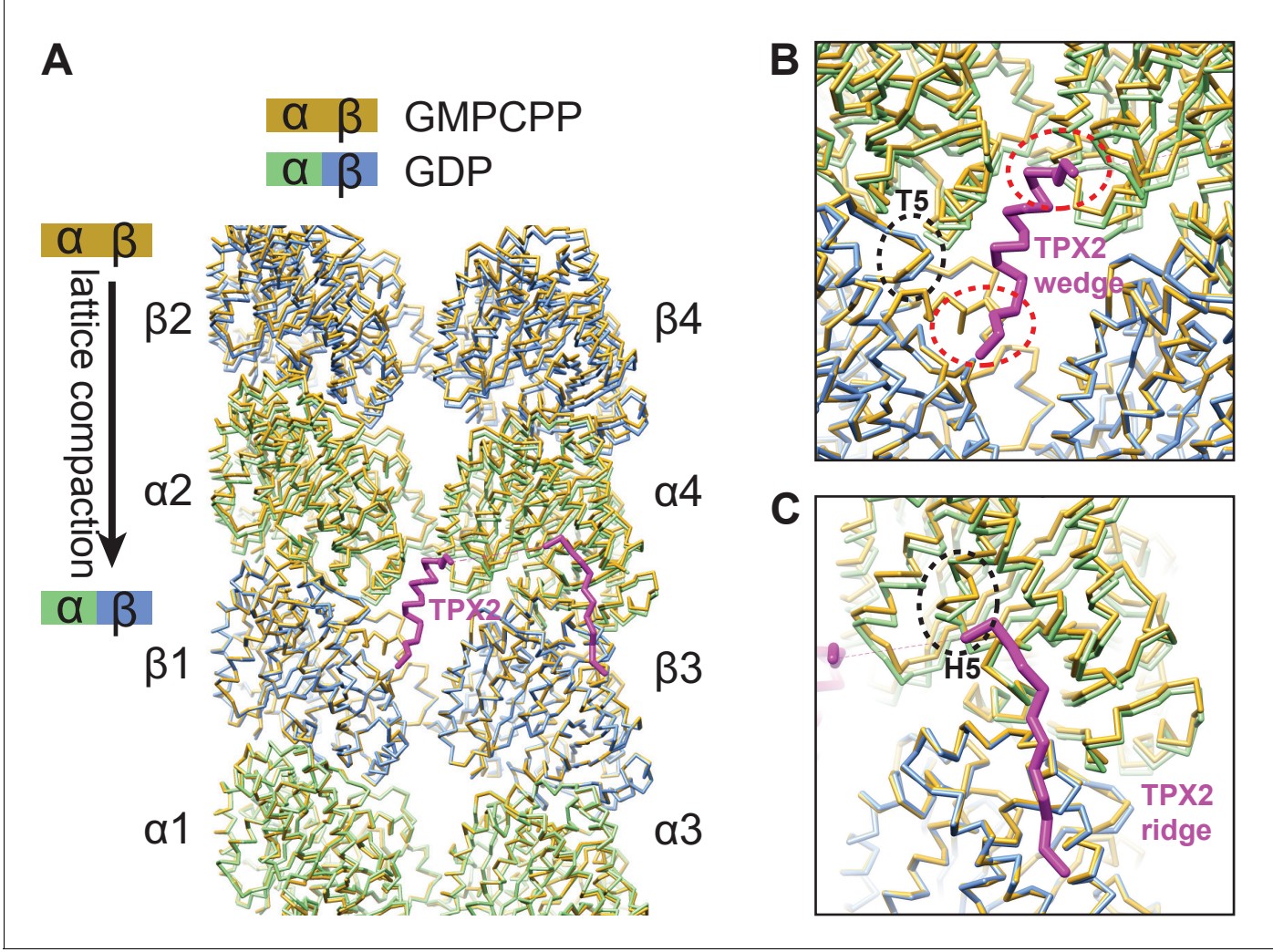

**Figure 5.** TPX2 binds at inter-dimer interfaces that change during the MT lattice compaction linked to GTP hydrolysis. (**A**) Comparison of atomic models between the kinesin-bound GMPCPP-MT and GDP-MT states (both in the absence of TPX2 binding). The two models are aligned on the β1-tubulin subunit. Both α- and β-tubulin in the GMPCPP-state are colored in orange, whereas α-tubulin and β-tubulin in the GDP-state are colored in green and blue, respectively. The model of the TPX2 molecule (magenta) from the present study is also displayed at the corresponding location. (**B**) Zoom-in view of the TPX2-wedge-binding site. The black dashed circle marks the T5 loop in β-tubulin that show significant local changes during MT lattice compaction. The red dashed circles mark the regions of tubulin contacting the short helix of the wedge element. (**C**) Zoom-in view of the TPX2-ridge binding site. The black dashed circle marks the H5 helix in α-tubulin that show significant local changes during MT lattice compaction.
DOI: https://doi.org/10.7554/eLife.30959.012

The following source data and figure supplements are available for figure 5:

**Figure supplement 1.** Plot of MT lattice parameters for different functional states.
DOI: https://doi.org/10.7554/eLife.30959.013

**Figure supplement 1—source data 1.** Lattice parameters for different MT states.
DOI: https://doi.org/10.7554/eLife.30959.014

**Figure supplement 2.** Effect of TPX2 binding on tubulin conformational transitions with nucleotide state.
DOI: https://doi.org/10.7554/eLife.30959.015

ridge bridges across the tubulin interdimer interface along the PF, TPX2 is also likely to stabilize PFs longitudinally. This effect on both lateral and longitudinal contacts might thereby provide a structural explanation for why TPX2 directly promotes MT formation and stability so efficiently.

Besides the ridge and wedge motifs, that our results have identified as critical contributors to TPX2-dependent MT nucleation, other parts of the TPX2 molecule can affect MT nucleation indirectly via other interaction partners such as Aurora A, γTuRC and augmin (*Alfaro-Aco et al., 2017*;

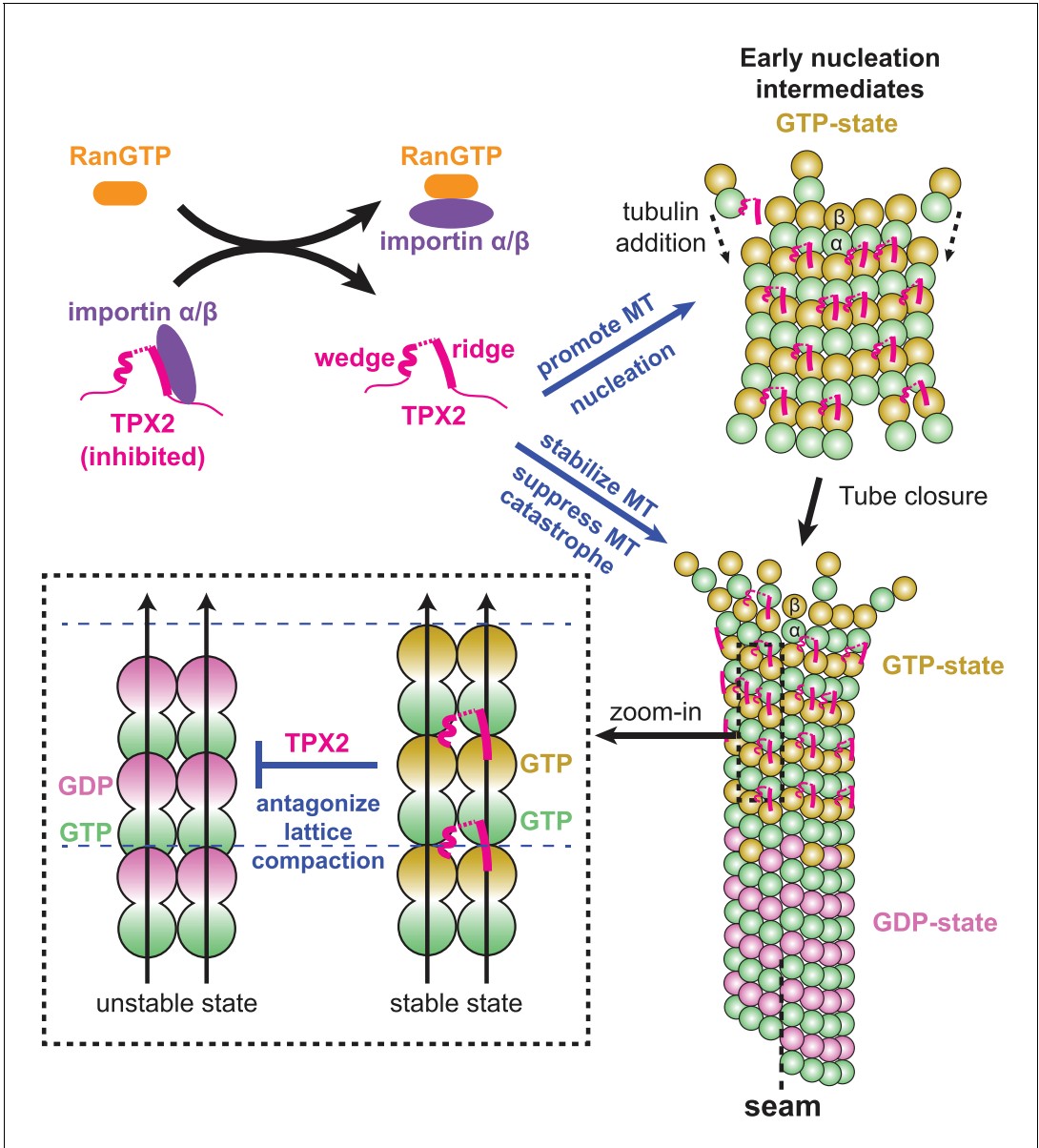

**Figure 6.** Model for the RanGTP-regulated interaction of TPX2 with tubulin assemblies during MT nucleation and MT growth. Binding of importins by RanGTP releases the sequestration of structural elements in TPX2 that are involved in interaction with tubulin assemblies at polymerization interfaces. TPX2 then functions to bring together tubulin subunits during MT nucleation and MT growth. The binding of TPX2 at those interfaces is optimal for the GTP-bound, extended MT lattice, and may slow down the GTP hydrolysis process by antagonizing MT lattice compaction.

DOI: https://doi.org/10.7554/eLife.30959.016

The following figure supplement is available for figure 6:

**Figure supplement 1.** Comparison of the binding sites of TPX2, EB3 and doublecortin (DCX) on MTs.

DOI: https://doi.org/10.7554/eLife.30959.017

*Bayliss et al., 2003*; *Pinyol et al., 2013*; *Scrofani et al., 2015*; *Tsai and Zheng, 2005*). There is increasing evidence that TPX2 enhances the activity of γTuRC-dependent MT nucleation either indirectly (*Pinyol et al., 2013*) or directly (*Alfaro-Aco et al., 2017*). In the light of the new binding mode of TPX2 observed here in the context of a MT, it is tempting to speculate that TPX2 may bind in a very similar manner to the interface between the γ-tubulin surface of γTuRC and the first tubulin layer, that is at the γ-tubulin/first α-tubulin interface, thereby directly promoting the templating

activity of γTuRC in addition to stabilizing interdimer tubulin-tubulin contacts in the MT wall, as discussed above. Future structural studies are required to explore this possibility.

### Regulation of TPX2 by importin binding

Our studies have identified two structural elements within TPX2, the ridge and the wedge, involved in specific interactions with the MT lattice. Our atomic model, validated using different constructs as well as mutational analysis, identified the sequences of these two MT-binding elements as overlapping extensively with the importin-α-binding site and the NLS of TPX2 (*Giesecke and Stewart, 2010*; *Schatz et al., 2003*) (*Figure 1—figure supplement 1*). Therefore, importin binding to TPX2 competes with MT binding of the central part of TPX2, explaining previous observations with purified proteins (*Roostalu et al., 2015*), preventing TPX2 from stabilizing MTs and from promoting their nucleation (*Figure 6*). Differences between the conservation of TPX2 sequences and its role in MT nucleation have been noted earlier (*Goshima, 2011*; *Karsenti, 2005*). The observation that the MT contact sites in the wedge and ridge of TPX2 appear to be conserved only in organisms, such as vertebrates, where a role of TPX2 for chromatin-dependent MT nucleation has been reported (*Gruss et al., 2001*; *Gruss et al., 2002*; *Vos et al., 2008*) and not where TPX2 does not seem to be involved in this pathway (*Hayward et al., 2014*; *Ozlü et al., 2005*) (*Figure 3—figure supplement 1*), suggests that this binding mode is distinctly responsible for Ran-GTP-regulated effects of TPX2 on MT nucleation and stability. Hence, our findings provide a mechanistic explanation of how the Ran-GTP gradient is coupled to MT nucleation and stabilization during mitosis.

### Conclusion

The present work sheds light on the process of regulated MT nucleation through a combination of the direct visualization of the MT stabilizer TPX2 bound to the MT surface and biochemical reconstitution assays that verify the functional importance of this interaction (*Figure 6*). Through a novel binding mode involving small, flexibly linked structural motifs, TPX2 binds across longitudinal and lateral interfaces between tubulin subunits in the MT lattice promoting the association between tubulin subunits (*Figure 6*, top right and bottom right). At the same time, TPX2-binding motifs allow the discrimination between nucleotide states in the MT lattice, possibly also slowing down GTP hydrolysis and the transition to a compacted GDP-MT lattice (*Figure 6*, bottom left). All these properties likely contribute to the direct effect TPX2 has on MT nucleation. The new binding mode might also contribute to stimulating γTuRC-mediated MT nucleation, in combination with indirect effects of TPX2 mediated by other interaction partners. By identifying the critical MT-binding regions in TPX2 as those that also bind importins, our study explains how the interaction of TPX2 with MTs is regulated by the Ran-GTP gradient. Similar molecular mechanisms may be shared among a group of nuclear proteins that are activated upon nuclear envelope breakdown and function as spindle assembly factors in mitosis and meiosis.

## Materials and methods

### Cloning and protein biochemistry

The mGFP-TPX2$^{mini}$ construct (containing residues 274–659 of human TPX2 N-terminally tagged with monomeric GFP) was described previously (*Roostalu et al., 2015*). The fusion protein was expressed in *Sf*21 cells and purified as described (*Roostalu et al., 2015*), concentrated to ~5 mg/ml with Vivaspin 15R concentrators (10,000 MWCO, Sartorius), ultracentrifuged (278,088 x *g*, 10 min, 4°C), and flash frozen and stored in storage buffer (50 mM HEPES (pH 7.5), 300 mM KCl, 2 mM MgCl$_2$, 50 mM arginine, 50 mM glutamate, 250 mM sucrose, 5 mM 2-mercaptoethanol (2-ME)) in liquid nitrogen.

To generate a bacterial expression construct for TPX2$^{micro}$, a fragment of the TPX2 cDNA encoding residues 274–370 was amplified by PCR using the TPX2$^{mini}$ construct as a template and cloned into a pETMZ vector together with the N-terminal mGFP resulting in a fusion His$_6$-Ztag-mGFP-Gly$_5$A-laMet-TPX2$^{274-370}$ where the His$_6$ and the Ztag could be cleaved off by TEV protease.

The TPX2$^{micro}$ fusion protein was expressed in *E. coli* BL21 pRil at 18°C for 16 hr, induced by 0.1 mM IPTG. To purify the protein, cell pellets from 2 l culture were resuspended in ice-cold lysis buffer (50 mM HEPES (pH 8.0), 300 mM KCl, 5 mM MgCl$_2$, 1 mM imidazole, 25 mM sucrose, 1 mM EDTA,

5 mM 2-ME) supplemented with a protease inhibitor cocktail (Roche) and DNAseI (10 μg/ml, Sigma Aldrich) and lysed using a microfluidizer. The lysate was clarified by ultracentrifugation (183,960 x *g*) for 45 min at 4°C and loaded on 2.5 g Protino Ni-TED resin (Macherey-Nagel). The resin-bound protein was washed with 40 ml of lysis buffer, 10 ml of lysis buffer containing 5 mM ATP, and then again with 40 ml lysis buffer. The protein was eluted by GST-TEV protease cleavage on a rotating wheel at 4°C for 16 hr. The soluble protein was then separated from the resin by centrifugation (700 x *g*, 10 min, 4°C). GST-TEV was then removed by 30-min incubation with glutathione resin (Novagen) on ice. The resin was then pelleted (700 x *g*, 10 min, 4°C) and the buffer of the mGFP-TPX2$^{micro}$ containing protein solution exchanged to MES A buffer (20 mM MES (pH 6.0), 2 mM MgCl$_2$, 5 mM 2-ME) via PD-10 columns (GE Healthcare). The protein was then loaded on a MonoS 5/50 GL column (GE Healthcare) pre-equilibrated with MES A buffer. The protein was eluted with an increasing linear KCl concentration gradient in MES A buffer. The peak fractions (eluted at ~330 mM KCl) were pooled, aliquoted and stored in liquid nitrogen.

To generate mutants, the expression construct of mGFP-TPX2$^{micro}$ was modified by PCR mutagenesis to disrupt either the 'ridge' (F307A or F307E) or the 'wedge' (F334A H335A or F334E H335E) regions, or both simultaneously. The expression and purification of the mutant proteins was carried out as described above for the mGFP-TPX2$^{micro}$. mGFP-TPX2$^{mini}$ and BAP-mTagBFP-TPX2$^{mini}$ (BAP – biotin acceptor peptide, BFP – blue fluorescent protein) triple mutants (F307E F334E H335E) were also generated by PCR mutagenesis, expressed in *Sf*21 cells and purified as described previously for wild-type mGFP-TPX2$^{mini}$ and BAP-mTagBFP-TPX2$^{mini}$ (*Roostalu et al., 2015*).

Porcine brain tubulin for total internal microscopy (TIRFM) assays was purified as described earlier (*Castoldi and Popov, 2003*) and labeled either with CF640R-*N*-hydroxysuccinimide ester (NHS, Sigma-Aldrich), Atto565-NHS ester (Sigma-Aldrich), or Alexa647-NHS ester or biotin-NHS ester (Thermo Scientific) according to established methods (*Hyman et al., 1991*). The porcine tubulin used for cryo-EM studies was purchased from Cytoskeleton (see below).

All new expression constructs were verified by sequencing. Protein concentrations were determined by Bradford assay (TPX2 constructs), or by measuring the absorbance at 280 nm (tubulin). TPX2 concentrations indicate monomer concentrations, tubulin concentrations refer to tubulin dimers.

## Total internal reflection fluorescence microscopy (TIRFM)

Flow chambers were assembled from a poly-(L-lysine)-polyethylene glycol (PLL-PEG, SuSoS) passivated counter glass and a biotin-PEG-functionalized coverslip as described previously (*Bieling et al., 2010*). TIRFM imaging was performed at 30 ± 1°C using either an iMIC system (FEI Munich) characterized in detail elsewhere (*Maurer et al., 2014*), or a custom TIRFM microscope (Cairn Research, Faversham, UK) based on a Nikon Ti-E frame with a 100 × 1.49 N.A. objective lens and with Andor iXon Ultra 888 EMCCD camera. The exposure times were always 150 ms at 1 s or 1.5 s intervals using either 488 nm (for mGFP), 561 nm (for Atto561), or 640 nm (for CF640R, or Alexa647) lasers for excitation for dynamic MT assays. Images were acquired with a 200 ms exposure time at 2 s intervals using a 638 nm laser for the nucleation assay (for CF640R). For double-color imaging images were acquired either simultaneously with 488 nm and 640 nm lasers, or alternating between 488 nm and 561 nm excitation to avoid bleed through. Image alignment was performed as described earlier using MATLAB (*Maurer et al., 2014*). Images were assembled and processed (image stabilization, background subtraction and generation, average Z-projections) using Fiji.

For MT dynamics assays, the GMPCPP-stabilized fluorescently labeled (containing either 12% of CF640R-, Alexa647- or Atto565-labeled tubulin) and biotinylated MT 'seeds' were polymerized as described earlier (*Bieling et al., 2010*; *Roostalu et al., 2015*). The assay itself was performed as previously described (*Roostalu et al., 2015*) with minor modifications. The passivated flow chambers were incubated first for 5 min with 5% Pluronic F-127 in MQ water (Sigma-Aldrich) at room temperature and then washed with assay buffer (for mGFP-TPX2$^{micro}$ AB: 80 mM PIPES, 1 mM EGTA, 1 mM MgCl$_2$, 1 mM GTP, 5 mM 2-ME, 0.15% (w/vol) methylcellulose (4000 cP, Sigma-Aldrich), 1% (w/vol) glucose, 0.02% (vol/vol), Brij-35; for experiments with mGFP-TPX2$^{mini}$ the AB also included 60 mM KCl) containing κ-casein (50 μg/ml, Sigma-Aldrich). The flow chamber was subsequently incubated on a metal block on ice in the same buffer additionally supplemented with NeutrAvidin (50 μg/ml, Life Technologies). Excess NeutrAvidin was then removed by washes with AB. Next the GMPCPP-'seeds' diluted in AB were flowed in and incubated in the chamber for 3 min at room temperature to

facilitate attachment. The unbound 'seeds' were removed by additional washes with AB followed by flowing in the final assay mix.

The final assay mix for experiments with TPX2$^{micro}$ proteins consisted of: 77.8% (vol/vol) AB containing mGFP-TPX2$^{micro}$ protein and 22.2% BRB80 (80 mM PIPES, 1 mM EGTA, 1 mM MgCl$_2$) containing oxygen scavengers (catalase and glucose oxidase), and fluorescently labeled tubulin (containing 5% of either CF640R-, Alexa647- or Atto565-labelled tubulin). The final protein concentrations were 500 nM or 1 μM for wild-type or mutant mGFP-TPX2$^{micro}$ proteins, 12.5 μM or 15 μM tubulin, and 180 μg/ml catalase (Sigma-Aldrich), 750 μg/ml glucose oxidase (Serva). To allow for direct comparisons the mGFP-TPX2$^{micro}$ and the respective mutants were all first pre-diluted to 50 μM in their storage buffer (20 mM MES (pH 6.0), 330 mM KCl, 2 mM MgCl$_2$, 5 mM 2-ME) prior further dilutions in AB and imaged on the same day under identical conditions. The final assay mix for experiments with TPX2$^{mini}$ proteins consisted of: 76.1% AB supplemented with 60 mM KCl and 22.2% BRB80 containing oxygen scavengers, and fluorescently labeled tubulin (containing 5% of either Alexa647-labelled tubulin), and 1.7% TPX2$^{mini}$ diluted in its storage buffer (see below). The final protein concentrations were 125 nM for wild-type or mutant mGFP-TPX2$^{mini}$ proteins, 12.5 μM tubulin, and 180 μg/ml catalase and 750 μg/ml glucose oxidase. To allow for direct comparisons between mGFP-TPX2$^{mini}$ and mGFP-TPX2$^{mini}$ F307E F334E H335E, the proteins were first diluted to 15 μM in their storage buffer (50 mM HEPES (pH 7.5), 300 mM KCl, 2 mM MgCl$_2$, 50 mM arginine, 50 mM glutamate, 250 mM sucrose, 5 mM 2-ME) and imaged on the same day under identical conditions.

TIRFM-based MT nucleation assays were performed as described previously (*Roostalu et al., 2015*). In short, the final assay mix (see below) was first prepared on ice and ultracentrifuged (278,088 × *g*, 7 min, 4°C). In parallel, the flow chamber was sequentially incubated for 10 min with 5% Pluronic F-127 at room temperature, washed with AB containing 60 mM KCl and 50 μg/ml κ-casein, and subsequently incubated on a metal block on ice in the same buffer additionally supplemented with NeutrAvidin (50 μg/ml). The flow chamber was then washed with AB containing KCl at room temperature and placed on a metal block at 30°C. The ultracentrifuged final assay mix was then incubated at 30°C for 1 min to initiate nucleation in solution and then transferred to the pre-warmed flow chamber. The chamber was sealed with silicone grease. Imaging was started 3 min after placing final assay mix at 30°C. The composition of the final assay mix: 80% AB containing 60 mM KCl, 18.7% BRB80 containing oxygen scavengers, bovine serum albumin (BSA, Sigma-Aldrich, fluorescently labeled tubulin (containing 5% of CF640R-labeled tubulin), and 1.3% wild-type or triple mutant BAP-mTagBFP-TPX2$^{mini}$ proteins in their storage buffer. The final protein concentrations were 90 nM for wild-type or triple mutant BAP-mTagBFP-TPX2$^{mini}$ proteins, 12.5 μM tubulin, 1 mg/ml for BSA, 180 μg/ml catalase and 750 μg/ml glucose oxidase. Three independent experiments were performed for the indicated condition.

## Fluorescence intensity analysis

All fluorescence intensity measurements were performed using Fiji software. To quantify the binding of mGFP-TPX2$^{micro}$ and mGFP-TPX2$^{mini}$ to the GMPCPP stabilized 'seed' part of the dynamic MT (*Figure 3F* and *Figure 4A*, respectively) a 50 pixel rolling-ball background subtraction was applied to each frame of a 250 timeframe movie. The fluorescence intensities were then averaged over all frames generating a single time-averaged image for the entire movie. The bright MT 'seeds' were identified and marked manually along their length with a three-pixel wide segmented line in the MT channel of the movie. The corresponding average mGFP-TPX2$^{micro}$ or mGFP-TPX2$^{mini}$ fluorescence intensities in these 'seed' areas were then determined in the GFP channel. To obtain the final average fluorescence intensity values, a residual background was subtracted. This residual background was generated from the time-averaged image by generating a 50-pixel rolling–ball background image followed by averaging its intensities. For mGFP-TPX2$^{micro}$ averaged 'seed' intensities for one sample were quantified. Number of 'seeds' measured: WT – 18, F307A – 30, F307E – 25, F334A H335A – 25, F334E H334 – 29, F307A F334A H335E – 29, F307E F334E H335E – 24. For mGFP-TPX2$^{mini}$ averaged 'seed' intensities for three samples were quantified. Number of 'seeds' measured: WT – 57, F307E F334E H335E – 67.

To quantify the average fluorescence intensities of mGFP-TPX2$^{mini}$ at growing MT plus ends (*Figure 4D*) averaged intensity profiles were measured similarly as described previously (*Roostalu et al., 2015*). In short, kymographs were generated of growing MTs. The growing plus

ends were then marked by three-pixel wide segmented lines. The kymographs were straightened, aligned and averaged together using the marked plus end as a reference point. The resulting image was then further averaged along the time axis (180 timeframes) to generate a time-averaged spatial intensity profile for the MT plus end. Number of plus ends analyzed: WT – 42, F307E F335E H335E – 43.

## Cryo-EM sample preparation

Porcine tubulin powder (Cytoskeleton) was reconstituted to 10 mg/ml in CB1 buffer (80 mM PIPES pH 6.8, 1 mM EGTA, 1 mM MgCl$_2$, 1 mM GTP, 10% glycerol). After one polymerization-depolymerization cycle, active tubulin was resuspended in cold EM buffer (80 mM PIPES, pH 6.8, 1 mM EGTA, 1 mM MgCl$_2$, 1 mM DTT, 0.05% Nonidet P-40) supplemented with 1 mM GMPCPP. GMPCPP-loaded tubulin at 3 mg/ml soluble tubulin concentration were polymerized at 37°C for about 1 hr, and GMPCPP-MTs were diluted to 0.25 mg/ml in warm EM buffer supplemented with 1 mM GMPCPP. mGFP-TPX2$^{mini}$ or mGFP-TPX2$^{micro}$ was desalted into cold EM buffer using a Zeba Micro Spin desalting column (Thermo Scientific), and the sample was clarified by ultracentrifugation at 80,000 RCF for 15 min at 4°C using a Beckman TLA-100 rotor. 3 µl GMPCPP-MT specimen was applied to a glow-discharged C-flat 1.2/1.3–4C holey carbon EM grid (Protochips). After 30 s incubation inside a Vitrobot (Maastricht Instruments) set at 15°C (to minimize protein aggregation) and 95% humidity, the grid was washed twice with 3 µl of 20 µM TPX2 (30 s incubation each time) to maximize the decoration of TPX2 on the MT lattice, before blotting and vitrification in liquid ethane.

## Cryo-EM data collection

The cryo-EM data for the mGFP-TPX2$^{mini}$ (initial testing) or mGFP-TPX2$^{micro}$ decorated GMPCPP-MT were collected using a 300 keV low-base Titan microscope (FEI) (located at UC Berkeley) with a K2 Summit direct electron detector (Gatan). The sample was imaged under parallel illumination conditions, with a beam diameter of ~2 µm on the specimen. A defocus range from −1.2 to −3.5 µm was used. All cryo-EM images were recorded at a nominal magnification of 27,500×, corresponding to a calibrated pixel size of 1.33 Å. The K2 camera was operated in counting mode, with a dose rate of ~8 electrons/pixel/s on the camera. Each exposure was 6 s long and recorded as a movie of 20 frames, corresponding to a dose of 1.37 electron/Å$^2$ for each frame, and an accumulative dose of 27.6 electrons/Å$^2$ on the specimen. The data were collected semi-automatically using the Leginon software suite (*Suloway et al., 2005*).

   A large cryo-EM dataset of the mGFP-TPX2$^{mini}$ decorated GMPCPP-MT was collected on a 300 keV Titan Krios microscope (FEI) at the HHMI Janelia Research Campus. The microscope is equipped with a spherical aberration corrector (Cs- correction) and a high-brightness field emission gun (X-FEG). A Gatan Image Filter (GIF) for energy filtering was used for data collection, with a slit width of 20 eV. A defocus range from −1 to −2.5 µm was used. A total number of ~4000 movie stacks were recorded on a post-GIF K2 Summit direct electron detector camera (Gatan), in super resolution mode with a calibrated pixel size of 1.35 Å per physical pixel and a dose rate of ~8 electrons/pixel/s. Each exposure was 7.5 s long and recorded as a movie of 25 frames, corresponding to a dose of 1.32 electron/Å$^2$ for each frame, and an accumulative dose of 33.0 electrons/Å$^2$ on the specimen. The data were collected automatically using SerialEM (*Mastronarde, 2005*).

## Image processing

For data collected at the HHMI Janelia cryoEM facility, each movie stack was subject to an anisotropic magnification correction using *mag_distortion_correct* (*Grant and Grigorieff, 2015a*), followed immediately by Fourier binning by 2. The calibrated pixel size after the correction and binning is also 1.33 Å. No significant magnification anisotropy (>0.5%) was detected for the Titan microscope located at UC Berkeley. Drift correction for each movie stack was performed using the UCSF *motioncorr* program(*Li et al., 2013*). Then the contrast transfer function (CTF) parameters were estimated from the motion-corrected micrographs using CTFFIND4 (*Rohou and Grigorieff, 2015*). Subsequently, we manually selected MTs from these motion-corrected micrographs using the APPION image processing suite (*Lander et al., 2009*). The MT selections were converted to overlapping boxes (512 × 512 pixels), with ~80 Å non-overlapping region (along the MT axis) between adjacent boxes. The initial alignment parameters and PF number for each boxed MT particle were

determined using multi-reference alignment (MRA) in EMAN1 (*Ludtke et al., 1999*). MT particles with the same PF number were merged and subject to further structural refinement in FREALIGN v9 (*Grigorieff, 2007*).

For high-resolution structural refinement in FREALIGN, we used the 'polished' particles obtained by *alignparts_lmbfgs* (*Rubinstein and Brubaker, 2015*), which tracks the movements for individual particles throughout the movie series, and applies dose exposure filtering (*Grant and Grigorieff, 2015b*). Starting from the initial alignment parameters obtained by EMAN1, and using a recently developed data processing protocol (*Zhang and Nogales, 2015*), we could accurately determine the α, β-tubulin register and seam location for each MT segment. This approach allows the study of MT-MAP interactions without the need for a large protein marker for the tubulin dimer (such as a kinesin motor domain, as we previously used *Alushin et al., 2014*; *Zhang et al., 2015*]), which could otherwise interfere with the binding of the MAP of interest. Finally, 3D reconstructions (assuming either pseudo-helical symmetry or no symmetry) were performed using the 'polished' particles, following a previously described protocol (*Zhang et al., 2015*; *Zhang and Nogales, 2015*). The final resolution for each reconstruction (*Figure 1—figure supplement 4A*) was estimated by calculating the Fourier Shell Correlation (FSC) of a single tubulin dimer from the odd and even maps, using a FSC 0.143 criterion. The local resolution (*Figure 1—figure supplement 4B,C*) was calculated using the *blocres* function in the Bsoft package (*Heymann and Belnap, 2007*).

### Atomic model building and refinement

The atomic models of TPX2 and α/β tubulin were built in COOT (*Emsley et al., 2010*), based on the high-resolution cryo-EM density map. The model for the two resolved structural elements of TPX2 was built de novo, while the tubulin model was built using our previous cryo-EM-derived structure of kinesin decorated GMPCPP-MT (PDB ID: 3JAT [*Zhang et al., 2015*]) as the starting point. Torsion angle, planar peptide and Ramachandran restraints were used during the building process in COOT.

The models built in COOT (TPX2 and α/β tubulin) were duplicated and fitted as a rigid-body into the MT lattice. And the initial model containing six tubulin dimers and two TPX2 molecules were subsequently refined with REFMAC v5.8 adapted for cryo-EM (*Brown et al., 2015*), following a previous protocol (*Zhang et al., 2015*). Secondary structure and reference restraints generated with ProSMART (*Nicholls et al., 2012*) were used throughout the refinement process. During refinement, local symmetry restraints were used to restrain corresponding interatomic distances in symmetry-related molecules. These local symmetry restraints are functionally analogous to non-crystallographic symmetry (NCS) restraints used during crystallographic refinement (*Murshudov et al., 2011*).

### Molecular graphics

All structural figures were generated using UCSF Chimera (*Goddard et al., 2007*; *Pettersen et al., 2004*).

### Data deposition

The following cryo-EM maps have been deposited in the Electron Microscopy Data Bank [EMDB]: TPX2$^{mini}$-decorated GMPCPP-MT (EMD-7101), TPX2$^{micro}$-decorated GMPCPP-MT (EMD-7102). The refined atomic model for TPX2$^{mini}$-decorated GMPCPP-MT has been deposited in the Protein Data Bank (PDB) with accession code 6BJC.

## Acknowledgements

We thank Rick Huang at the HHMI Janelia CryoEM Facility for help in data collection. We thank Claire Thomas for insect cell culture maintenance and baculovirus generation and Nicholas I Cade for help with kymograph data analysis. We are thankful to Patricia Grob and Abhiram Chintangal for EM and computer support, and to Basil Greber for assistance in model building. This work was funded by a grant from NIGMS (GM051487 to EN). TS and JR are supported by the Francis Crick Institute, which receives its core funding from Cancer Research UK (FC001163), the UK Medical Research Council (FC001163), and the Wellcome Trust (FC001163). JR was also supported by a Sir Henry Wellcome Postdoctoral Fellowship (100145/Z/12/Z) and TS acknowledges support from the European Research Council (Advanced Grant, project 323042). EN is a Howard Hughes Medical Institute investigator.

## Additional information

### Funding

| Funder | Grant reference number | Author |
| --- | --- | --- |
| Wellcome Trust | Sir Henry Wellcome Postdoctoral Fellowship 100145/Z/12/Z | Johanna Roostalu |
| Cancer Research UK | FC001163 | Thomas Surrey |
| Medical Research Council | FC001163 | Thomas Surrey |
| FP7 Ideas: European Research Council | Advanced Grant 323042 | Thomas Surrey |
| Wellcome Trust | FC001163 | Thomas Surrey |
| National Institute of General Medical Sciences | GM051487 | Eva Nogales |
| Howard Hughes Medical Institute | | Eva Nogales |

The funders had no role in study design, data collection and interpretation, or the decision to submit the work for publication.

### Author contributions

Rui Zhang, Conceptualization, Data curation, Software, Formal analysis, Validation, Investigation, Visualization, Methodology, Writing—original draft, Project administration, Writing—review and editing; Johanna Roostalu, Conceptualization, Resources, Formal analysis, Funding acquisition, Validation, Investigation, Visualization, Methodology, Writing—original draft, Project administration, Writing—review and editing; Thomas Surrey, Eva Nogales, Conceptualization, Resources, Supervision, Funding acquisition, Validation, Writing—original draft, Project administration, Writing—review and editing

### Author ORCIDs

Thomas Surrey https://orcid.org/0000-0001-9082-1870
Eva Nogales https://orcid.org/0000-0001-9816-3681

### Decision letter and Author response

Decision letter https://doi.org/10.7554/eLife.30959.025
Author response https://doi.org/10.7554/eLife.30959.026

## Additional files

### Supplementary files

• Transparent reporting form
DOI: https://doi.org/10.7554/eLife.30959.018

### Major datasets

The following datasets were generated:

| Author(s) | Year | Dataset title | Dataset URL | Database, license, and accessibility information |
| --- | --- | --- | --- | --- |
| Rui Zhang, Johanna Roostalu, Thomas Surrey, Eva Nogales | 2017 | TPX2_mini decorated GMPCPP-microtubule | https://www.rcsb.org/pdb/explore/explore.do?structureId=6BJC | Publicly available at the RCSB Protein Data Bank (accession no. 6BJC) |

| Rui Zhang, Johanna Roostalu, Thomas Surrey, Eva Nogales | 2017 | TPX2_mini decorated GMPCPP-microtubule | https://www.ebi.ac.uk/pdbe/entry/emdb/EMD-7101 | Publicly available at the Electron Microscopy Data Bank (accession no. EMD-7101) |
|---|---|---|---|---|
| Rui Zhang, Johanna Roostalu, Thomas Surrey, Eva Nogales | 2017 | TPX2_micro decorated GMPCPP-microtubule | https://www.ebi.ac.uk/pdbe/entry/emdb/EMD-7102 | Publicly available at the Electron Microscopy Data Bank (accession no. EMD-7102) |

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
