## [Decision Letter]

Thank you for submitting your article "Structural Insight into TPX2-Stimulated Microtubule Assembly" for consideration by *eLife*. Your article has been favorably evaluated by Anna Akhmanova (Senior Editor) and three reviewers, one of whom is a member of our Board of Reviewing Editors. The following individual involved in review of your submission has agreed to reveal his identity: Torsten Wittmann (Reviewer #3).

The reviewers have discussed the reviews with one another and the Reviewing Editor has drafted this decision to help you prepare a revised submission.

Summary:

The manuscript by Zhang et al. provides a convincing structural and functional model into the regulation of microtubule stability by Tpx2. This is important as Tpx2 is an important player in Ran-mediated spindle assembly and has emerged as a marker of aggressive cancer phenotypes. Tpx2 also appears to bind to a large number of other proteins to stabilize and nucleate microtubules and contribute to MT branching. However the molecular basis for its function remains poorly understood.

The authors use cryo-EM to show how two peptides (the "wedge" and the "ridge") of Tpx2 interact both laterally and longitudinally along the microtubule, stabilizing key sites for microtubule polymerization. Furthermore, the TPX2 is shown to extend the microtubule lattice, counteracting the contraction observed upon GTP hydrolysis. These data paint a strong mechanistic argument as to how TPX2 stabilizes microtubules in the cell, through making and maintaining contacts between subunits during growth, and promoting the more stable (GTP-bound) form of microtubule. The interactions are validated by mutagenesis and TIRF assays. Finally, the authors show that part of the region to interact with the microtubule also binds importin a/b, creating a simple and convincing explanation as to how the interaction can be regulated in the spindle as part of the RanGTP cycle.

Overall the paper is well written, interesting and easy to follow. The manuscript presents its arguments well, and provides a lot of new insight into an important part of spindle formation. The high-resolution of the reconstruction, and the accompanying structure-based mutagenesis, make the arguments particularly convincing. The manuscript is of broad interest and importance, and should be accepted after revisions.

Essential revisions:

1) In Figure 3, the authors show that the mutations decrease the binding of TPX2 to the microtubule. Can the authors provide data that would demonstrate a concurrent decrease in microtubule stability? For example, an increase in the rate of catastrophe, slower polymerization, or lower nucleation threshold in the presence of the mutant compared to WT.

2) The authors claim in the first paragraph of the "Conformational sensitivity of the TPX2 MT binding mode" Results section that TPX2 "appears to interact also with the C-terminal tail ("E-hook") of α-tubulin". In the corresponding figure (3A) there is nothing to demonstrate how the authors came to this conclusion. This point is also raised again in the Discussion. Is there any electron density that would help support this point? Or could the authors provide their reasoning for the claim.

---

## [Author Response]

Essential revisions:1) In Figure 3, the authors show that the mutations decrease the binding of TPX2 to the microtubule. Can the authors provide data that would demonstrate a concurrent decrease in microtubule stability? For example, an increase in the rate of catastrophe, slower polymerization, or lower nucleation threshold in the presence of the mutant compared to WT.

To test whether reduced binding of mutated TPX2 to the microtubule decreases the stimulation of microtubule nucleation by TPX2 compared to the wild type protein, we performed nucleation assays as previously described in Roostalu et al., NCB, 2015. We made new expression constructs for the production of triple-mutated biotinylated TPX2mini, expressed this protein in insect cells and purified it (Figure 1—figure supplement 5). In the previously developed nucleation assay, biotinylated TPX2 mediated the binding of microtubules nucleating in solution to functionalised, NeutrAvidin-coated glass where they could be observed by TIRF microscopy. Using this assay, we demonstrated now that microtubule nucleation was considerably reduced in the presence of the triple mutant of TPX2mini compared to wildtype TPX2mini (new Figure 4), in agreement with its reduced binding to growing microtubule ends and GMPCPP microtubules. These results demonstrate the functional importance of the identified binding site for the microtubule nucleating effect of TPX2 in vitro.

2) The authors claim in the first paragraph of the "Conformational sensitivity of the TPX2 MT binding mode" Results section that TPX2 "appears to interact also with the C-terminal tail ("E-hook") of α-tubulin". In the corresponding figure (3A) there is nothing to demonstrate how the authors came to this conclusion. This point is also raised again in the Discussion. Is there any electron density that would help support this point? Or could the authors provide their reasoning for the claim.

We agree with this criticism. We have reanalysed our data by filtering several cryoEM maps to the same resolution (3.5 Å) with the same B-factor sharpening (B factor -125). We observed that our cryo-EM structure of TPX2-GMPCPP-MT does not resolve significantly more density of α-tubulin C-terminal tail than other MT states we have determined by high-resolution cryo-EM, including undecorated and kinesin decorated MTs (Zhang et al., Cell, 2015 and unpublished data). We agree that our proposal was too speculative and we have removed the claim from the Results. This does not affect any of our main conclusions of the manuscript. In the Discussion of the manuscript, we have discussed how other binding sites in TPX2 may contribute to its microtubule stabilizing effect. We continue to think that this discussion is valuable.